# Metagenome-Based Analysis of the Microbial Community Structure and Drug-Resistance Characteristics of Livestock Feces in Anhui Province, China

**DOI:** 10.3390/vetsci11020087

**Published:** 2024-02-12

**Authors:** Ying Shao, Zhao Qi, Jinhui Sang, Zhaorong Yu, Min Li, Zhenyu Wang, Jian Tu, Xiangjun Song, Kezong Qi

**Affiliations:** 1Anhui Province Engineering Laboratory for Animal Food Quality and Bio-Safety, College of Animal Science and Technology, Anhui Agricultural University, Hefei 230036, China; shaoying@ahau.edu.cn (Y.S.); sangjhui@stu.ahau.edu.cn (J.S.); zhaorongyu@stu.ahau.edu.cn (Z.Y.); limin0410@stu.ahau.edu.cn (M.L.); wangzhenyu@ahau.edu.cn (Z.W.); tj-0531@ahau.edu.cn (J.T.); 2Anhui Province Key Laboratory of Veterinary Pathobiology and Disease Control, College of Animal Science and Technology, Anhui Agricultural University, Hefei 230036, China; 3School of Information and Artificial Intelligence, Anhui Agricultural University, Hefei 230036, China; qizhao1050@ahau.edu.cn

**Keywords:** metagenome, microbial resistance, virulence genes, swine

## Abstract

**Simple Summary:**

In this study, we collected swine feces at different physiological stages for macrogenomic sequencing. By comparing the sequencing results of the samples, we analyzed the differences in microbial species in these samples at different physiological states. In addition, we also compared the differences in drug-resistance genes among different samples in order to provide a basis for the development of scientific pig breeding policies and management measures.

**Abstract:**

We analyzed metagenome data of feces from sows at different physiological periods reared on large-scale farms in Anhui Province, China, to provide a better understanding of the microbial diversity of the sow intestinal microbiome and the structure of antibiotic-resistance genes (ARGs) and virulence genes it carries. Species annotation of the metagenome showed that in the porcine intestinal microbiome, bacteria were dominant, representing >97% of the microorganisms at each physiological period. Firmicutes and Proteobacteria dominated the bacterial community. In the porcine gut microbiome, the viral component accounted for an average of 0.65%, and the species annotation results indicated that most viruses were phages. In addition, we analyzed the microbiome for ARGs and virulence genes. Multidrug-like, MLS-like, and tetracycline-like ARGs were most abundant in all samples. Evaluation of the resistance mechanisms indicated that antibiotic inactivation was the main mechanism of action in the samples. It is noteworthy that there was a significant positive correlation between ARGs and the total microbiome. Moreover, comparative analysis with the Virulence Factor Database showed that adhesion virulence factors were most abundant.

## 1. Introduction

Antibiotics are one of the most important human discoveries of the 20th century [1]. However, in the process of livestock and poultry breeding, antibiotics have been frequently and irregularly used as a disease treatment and growth promoter, and the abuse phenomenon is very serious [2]. Along with the extensive use of antibiotics, the residual antibiotic parent and its metabolites enter water, sediment, soil, and other environmental media with livestock excreta, which promotes the enrichment of antibiotic-resistance genes (ARGs) in the environment, which constitutes a direct threat to staff engaged in aquaculture, as well as to residents living around the farms [3]. In addition, drug-resistant genes that enter the environment can also be transmitted to human-associated microorganisms and even pathogenic microorganisms through horizontal gene transfer (HGT), such as conjugation, transformation, and transduction. Some studies have shown that drug-resistant genes can also enter the human food chain by contaminating planted vegetables with manure, which triggers serious bacterial resistance, threatens the quality and safety of animal products and public health, and poses a great potential danger to human and animal health [4]. Numerous studies have shown that livestock and poultry manure from farms has become a new “reservoir” of drug-resistant bacteria and antibiotic-resistance genes (ARGs) in the natural environment. Pan et al. [5] found that the detection rate of tetracycline-resistance genes was as high as 84% in the manure from 21 farms. Brooks et al. [6] isolated several pathogen strains from cesspools on large-scale pig farms using fluorescence quantitative polymerase chain reaction (PCR). These authors also isolated several strains of pathogenic bacteria from fecal ponds of large-scale pig farms. They found that these strains carried ermA, ermF, tetA, tetB, and mecA, determined by fluorescence quantitative PCR. He et al. [7] examined ARGs in the feces of pigs from three commercial pig farms in southern China. They detected 22 ARGs with a high abundance; the detection rate of sul and cml ARGs reached 100% [8]. At present, China has one of the most serious problems related to drug-resistant bacteria of animal origin. Because the ARGs are diverse and widely distributed, the current situation is not optimistic [9].

In addition, the presence of virulence genes has been reported in pathogenic bacteria and even in some probiotics. Virulence genes, which enable microorganisms to colonize the surface of a host [10] or within the host to enhance their own pathogenicity [11], mainly include bacterial toxins, cell surface proteins that mediate bacterial attachment, cell surface carbohydrates and proteins that protect the bacteria [12], and hydrolytic enzymes that may contribute to bacterial pathogenicity [13]. Transmission of virulence genes among microorganisms can turn non-pathogenic bacteria into pathogenic ones. Toxicity genes can also contaminate soil, water, crops, and the food chain through feces, posing a threat to ecosystems and human health. Therefore, monitoring the distribution of virulence genes in livestock and poultry feces is essential for disease control and prevention.

In the past, culture-dependent microbiological methods have been the dominant approach used to study microorganisms, but this approach is inadequate when the study subject is environmental microorganisms [14]. This is because more than 99% of microorganisms in environmental samples cannot be isolated or are difficult to isolate and culture [15]. The advent of macrogenomic technology has revolutionized the field of microbiology. Compared with 16S rRNA gene sequencing, macrogenome sequencing can provide more accurate species classification, and based on macrogenome sequencing data, it can also be used to study the genes in the genome, which can directly reflect the biological functions of microorganisms [16]. In macrogenomics, the limitations of culture methods are overcome by directly extracting DNA from environmental samples, providing scientists with more comprehensive and accurate information about microorganisms. By sequencing and analyzing the total DNA in the samples, it is possible to identify the various groups of microorganisms within them, and to delve deeper into their genetic information, functions, and metabolic pathways [17]. Therefore, macrogenomic technologies have facilitated the study of the functionality of microbial communities. By analyzing the metabolic pathways and genomic functions of these microbes, scientists are able to better understand their roles in ecosystems, including beneficial ecological functions (e.g., organic matter degradation and nitrogen cycling), as well as their impact on human health and the environment. Overall, the emergence of macrogenomic technologies has not only accelerated the field of microbiology, but also expanded our understanding of microbial diversity and function in nature, providing strong support for solving environmental problems and developing biotechnology [18].

Currently, pork is one of the major consumed meat products, and there are fewer reports exploring pathogenic microorganisms and ARGs in untreated composted manure and the surrounding soil of pig breeding farms in Anhui Province. In this study, therefore, we analyzed the feces from sows at different physiological periods reared on large-scale pig farms in Anhui Province, China, using macrogenome sequencing and the NCBI and CARD (The Comprehensive Antibiotic Resistance Database) databases [19]. We aimed to obtain information on the distribution of the bacterial flora, virulence genes, and ARGs, and their distribution and abundance in the feces of sows during different physiological periods. This information could provide a basis for the development of scientifically based pig breeding policies and management measures.

## 2. Materials and Methods

### 2.1. Sample Collection

In this study, fecal samples were collected from sows at a large-scale farm in Anhui Province, China, Before sample collection, the hosts were fed and cultured in accordance with the relevant breeding specifications and immunization procedures of this large-scale farm, during the reserve period (BG), the vacant period (HB), the gestation period (SC), and the mating period (XZ). No fewer than three samples (each no less than 10 g) were collected for each period. All samples were placed on dry ice for low-temperature preservation. The samples were transported to the laboratory and preserved in a refrigerator at –80 °C in a timely manner. The samples were sterilized and cleaned up in a timely manner to avoid cross-contamination between samples. All samples were numbered and the background of the collection was recorded.

### 2.2. DNA Extraction, Library Construction, and Metagenome Sequencing

Total genomic DNA was extracted from the collected samples. The concentration and quality of the extracted total DNA were evaluated with 1% agarose gel electrophoresis. The genomes that passed the test were randomly fragmented to a size of about 400 base pairs (bp) using a Covaris^®^ M220 (Woburn, MA, USA) focused ultrasonic crusher, and PE libraries were constructed using the NEXTFLEX™ Rapid DNA-Seq Kit (Waltham, MA, USA). After the libraries passed the test, the samples were sequenced using the Illumina sequencing platform. During the sequencing process, multiple samples were sequenced in parallel. An index tag sequence was added to each sequence to indicate the source information of the sample. The data from each sample were distinguished according to the index sequence, and the extracted data were saved in the fastq format.

### 2.3. Assembly and Annotation of Metagenome Sequencing Data

Data analysis started from the downstream raw sequences. First, Fastp (v0.20.0) was used to quality check the raw sequences. BWA (v0.7.9a) was used for optimization such as de-hosting contamination. Megahit (v1.1.2) was used to splice and assemble the optimized sequences. Prodigal (v2.6.3) was used for open reading frame (ORF) prediction of contigs based on the splicing results. Kraken2 (John Hopkins University, Baltimore, MA, USA) with default parameters was used to annotate them.

### 2.4. Construction of Non-Redundant Gene Set

The overall information of all the genes in the environment can be described by constructing a non-redundant gene set; that is, the gene sequences predicted from all the samples are clustered by CD-HIT software (v4.6.1), and the longest gene in each class is taken as the representative sequence to construct the non-redundant gene set, so that the commonalities and differences between different samples can be explored. In this step, the results that can be obtained are the statistical table of the number and length of genes before and after de-redundancy, the base sequences of the genes in the non-redundant gene set, and the amino acid sequences of the genes in the non-redundant gene set.

### 2.5. Individualized Analysis

Gene-based species taxonomic annotation compares the NR database (v20200604) to obtain species and abundance information at each taxonomic level (domain, kingdom, phylum, order, order, family, genus, species) in each sample. This can be used as a basis for subsequent statistical analysis at the species level and further analysis of the samples using relevant dataset websites such as CARD (v3.0.9) and VFDB (the Virulence Factor Database) (v2020.07.03).

## 3. Results

### 3.1. Metagenome Sequencing Results and Alpha Diversity

We performed metagenome sequencing on the 12 collected samples. There were at least 40,000,000 raw reads from the sequencing of each sample. The N50 of the 12 samples was 715–1051 bp after the assembly was completed (Table 1). We predicted ORFs for the contigs with a nucleic acid length ≥100 bp in the splicing results of these 12 samples. There was an average of 477,706 ORFs for each sample. The data met the quality requirements for subsequent analysis. We combined the three samples in the same physiological state into one group, which ultimately produced four groups of samples (BG, HB, SC, and XZ). We annotated a total of 26,889 bacterial species after combining these four sets of samples to remove redundancy. However, it is worth noting that >18% of these bacteria could not be accurately categorized to the species level. These bacteria perhaps represent unidentified new species in the porcine gut.

To investigate the diversity of microorganisms in the samples of the dataset, we calculated the Shannon, Simpson, and Chao1 indices (Figure 1). The Chao1 index reflects the species richness of the communities in the samples, and the Simpson and Shannon indices estimate the diversity of the communities. Among them, the Simpson index is inversely proportional to diversity and the Shannon index is positively proportional to diversity. The alpha diversity box plot shows that the diversity and richness of microorganisms in the fecal samples of the four groups differed somewhat. In general, the SC group had the highest species diversity and the XZ group had the lowest diversity.

### 3.2. Beta Diversity

We performed principal coordinates analysis (PCoA) based on Bray–Curtis dissimilarity for the overall data. Each group showed obvious aggregation trends. In addition, the XZ, HB, and BG groups showed an aggregation trend (Figure 2), indicating that there was some similarity in their community composition. However, the SC group was at a certain distance from each group, possibly due to the distribution of rare species within this group.

### 3.3. Microbial Community Structure

We annotated the species to 5 domains, 216 phyla, 707 orders, 1293 families, 4326 genera, and 26,888 species. As shown in Figure 3, at the domain classification level, fecal microorganisms were dominated by bacteria. The relative abundance of bacteria in each sample ranged from 97.31% to 99.05%, with an average of 97.99%. The average relative abundance of the other groups was 0.32% for archaea, 0.65% for viruses, and 0.16% for fungi (Table 2).

### 3.4. Bacterial Community Composition

We found 165 bacterial phyla common to all samples. The abundance of the phyla varied among the samples (Figure 4a). Proteobacteria was the dominant phylum in the BG, XZ, and SC groups, accounting for 49.4%, 52.2%, and 66.6% of the microbial abundance in each group, respectively, while its abundance was only 20.9% in the HB group. Firmicutes was the most abundant phylum in the HB group, accounting for 43.1% of the microbiota; its content in the BG, XZ, and SC groups was 25.1%, 21.6%, and 10.2%, respectively. Bacteroidetes was the third most abundant phylum in the four groups. Overall, the relative abundance of the Firmicutes, Proteobacteria, and Bacteroidetes phyla accounted for >85% of the total number of microorganisms in the four groups. In terms of species composition at the genus level (abundance percentage >0.01%) (Figure 4b), the top three genera in the HB group were *Acinetobacter* (11.54%), *Prevotella* (8.14%), and *Lactobacillus* (5.94%). In the SC group, the most abundant genera were *Pseudomonas* (14.79%) and unclassified_c_Gammaproteobacteria (12.01%). *Pseudomonas* (29.4%) was the dominant genus in the XZ group, followed by unclassified_c_Gammaproteobacteria (9.1%). Finally, *Escherichia* (14.14%) was the most abundant genus in the BG group, followed by *Pseudomonas* (8.22%).

There were a total of 2643 bacteria at the same genus level for the four groups. The SC group had the most unique species (299) and the XZ group had the fewest unique species (62); the HB and BG groups had 126 and 129 unique species, respectively. These data suggest that the samples from the SC group had the highest species richness (Figure 4c).

We also analyzed the contribution of species to ARGs in greater depth. Figure 4d shows that the contribution of species to ARGs is positively related to the relative abundance. However, it is worth noting that the trend of the contribution of the same species in the same group to different ARGs is roughly the same, which suggests that perhaps most of the ARGs are contained in the dominant strains.

### 3.5. Viral Community Composition

Based on species annotation in the NR database, there were a total of 28 families and 3 unclassified families of viruses from the fecal microbiome. The families with an average relative abundance >0.1% are shown in Figure 5. *Siphoviridae* was the most abundant in all groups (except HB). The abundance of *Podoviridae*, *Myoviridae*, and *Drexlerviridae* varied greatly among the four groups. The distribution of these three families was relatively higher in the XZ and SC groups, and the distribution of *Myoviridae* was slightly lower than that of *Siphoviridae* in the BG group, with a relative abundance of 37.2%. The relative abundance of *Myoviridae* in the BG group was slightly lower than that of *Siphoviridae* at 37.09%, and the relative abundance of *Podoviridae* and *Drexlerviridae* in the HB group was 39.45% and 33.38%, respectively. Overall, most of the annotated viruses were phages that infect bacteria.

### 3.6. ARGs

The ARGs showed differences depending on the physiological period. By comparing these 12 metagenome sequencing samples with the CARD database, we identified a total of 60,710 genes coding for proteins identified as resistant, which were categorized into 872 different subtypes of ARGs (Appendix A). The SC group had the largest number of ARG subtypes (788) and the HB group had the lowest number (708) (Figure 6a). The SC group contained the most ARGs (35) (Figure 6b). ARGs could be further categorized into 21 classes depending on their target antimicrobial drug. Twenty of these ARG classes occurred in each group, while the fusidic acid class only occurred in the XZ group.

Overall, multidrug-like ARGs accounted for >30% of the abundance in all groups, followed by MLS-like ARGs and tetracycline-like ARGs (Figure 6c). Among the different ARG isoforms, glycopeptides (van isoforms) had the highest number of species, with 70, followed by tetracycline (tet), 23S ribosomal ribonucleic acid methyl methyltransferase (erm), aminoglycosides (AAC), and β-lactams (OXA). The major resistance mechanisms of ARGs were antibiotic efflux, antibiotic inactivation, antibiotic target alteration, antibiotic target protection, antibiotic target substitution, and reduced cell membrane permeability to antibiotics. Among these, antibiotic inactivation was the most common mechanism of action, while <1% of ARGs employed reduced cell membrane permeability to antibiotics as their mechanism of action (Figure 6e).

We performed the Mantel test to calculate the correlation between ARGs and microbes at the species level. There was a significant positive correlation between the total resistance genome and the total microbiome (mantel_r = 0.955, *p* = 0.001).

### 3.7. Microbial Virulence Factors

Through comparative analysis with the Virulence Factor Database (VFDB) (Figure 7), a total of 446 virulence factor genes were annotated (191 core and 255 predicted). Each group showed a similar distribution trend of the detected virulence factor types. The attack-type virulence factors accounted for the largest share of each group (approximately 50%), followed by non-specific virulence factors and defense virulence factors. Further analysis revealed that adhesion-related genes were the most prevalent among all types of virulence factors, followed by iron uptake system, secretion system, antiphagocytosis, regulation, toxin, invasion, serum resistance, stress protein, phase variation, protein, and magnesium uptake, each of which accounted for >1% of each sample. In addition, exoenzyme- and immunoglobulin-protease-related genes were also annotated.

## 4. Discussion

The main objective of this study was to characterize the functional content of the porcine fecal microbiome. Our comparative macrogenomic approach revealed taxonomic and functional elements of fecal microorganisms in sows at different physiological periods [20]. Alpha diversity analysis of the macrogenomic sequencing results revealed differences in the four groups of feces from sows at different physiological periods. The SC group had the highest Chao1 index, indicating that it had the highest number of species varieties among the four groups. The SC group also had the highest Shannon index, indicating greater community diversity. On the contrary, the Simpson index was inversely correlated to the microbial community diversity: the larger the Simpson index, the lower the community diversity. The XZ group obviously had the lowest microbial diversity of the four groups. The lower species diversity in the feces of mating sows relative to gestating sows may be due to the fact that to ensure successful gestation, the mating sows received antibiotics, and the use of antibiotics could have directly reduced microbial abundance. On the other hand, gestating sows show a hormonal surge, an increase in local circulation, and changes in dietary and behavioral habits that could underlie the increased microbial diversity. Some studies have shown that pigs with longer farrowing and lactation periods are at higher risk of acquiring resistance [21]. The changes in behavior during pregnancy affect the microbial community to varying degrees, and the discontinuation of antibiotics and other drugs may also have an effect on the microbial community response, which may be one of the reasons for the changes in gut microbial abundance in gestating sows [22]. The beta diversity analyses also showed that the species diversity of gestating sows was higher than that of the other groups, a phenomenon that may also be related to the discontinuation of antibiotics and the change in dietary habits. In addition, many environmental factors have been implicated in microbial resistance [21,23].

In recent decades, the threat of novel antibiotic resistance has emerged and spread globally [24]. As one of the world’s largest producers and consumers of food animals, China is facing a public health and ecological crisis due to the misuse of veterinary antibiotics in the production of animal food. Various antimicrobial drugs and residues have been found in animal feces in China; the most common are fluoroquinolones, followed by sulfonamides, tetracyclines, and MLS [25]. In general, the production of major ARGs is usually considered to be associated with antibiotics applied in human medicine and veterinary therapy, and for livestock weight gain [26]. The antibiotic inactivation mechanism was the primary mechanism of action for the development of resistance in the samples of this study, followed by the antibiotic efflux system. This is slightly different from previous findings [27], perhaps due to regional differences. There is a relationship between the different types of antibiotics to which animals are exposed during the farming process. The mechanism of antibiotic inactivation is mainly mediated by inactivating and passivating enzymes, which can hydrolyze or modify antimicrobial drugs to make them inactive before they reach their targets. Common enzymes are β-lactamases and aminoglycoside passivating enzymes, among others. Active exocytosis is a common mechanism by which bacteria can actively expel antimicrobials from the membrane, thus evading the action of antimicrobials. It is important to note that many active efflux systems are non-specific and therefore lead to the development of multi-drug resistance.

Antibiotics play a pivotal role in swine farming, being incorporated into feed or water to prevent and treat diseases in animals while simultaneously promoting growth. However, this widespread practice introduces antibiotics and antibiotic-resistant bacteria into swine feces, subsequently contaminating surface and groundwater, and thereby posing a significant public health hazard [28]. We analyzed the collected samples and found a total of 872 different subtypes of ARGs in these 12 samples, among which the content of multidrug ARGs and MLS-like ARGs was the highest. These ARGs possess a more complex composition. Tetracycline-like ARGs were the second most abundant after multidrug-like ARGs; these findings are closely related to the clinical use of antibiotics [29]. It is noteworthy that species contribution analysis revealed that some common resistance genes are present in most of the dominant flora. This phenomenon is inextricably linked to the long-term misuse of antibiotics in food animal farming, whether it is long-term sub-dose medication for disease prevention purposes or non-adherence to the administered regimen for economic purposes. Notably, the authors of a previous study found a significant positive correlation between the total resistance genome and the total microbiome, which demonstrates that ARGs are likely to be closely associated with a certain type of bacteria [30]. These related phenomena contribute to the proliferation of drug-resistant bacteria and the spread of ARGs.

We also compared the sequencing results with the VFDB database. There was a total of 446 virulence factor genes in the four groups, including 191 core virulence factors and 255 predicted virulence factors. Each group showed a similar distribution trend of virulence factor species, which suggests that the distribution of virulence factors may not vary much in different bacterial species.

## 5. Conclusions

We collected 12 fecal samples from sows in different physiological periods to analyze the differences in their gut microbiota. Bioinformatics analysis revealed the highest microbial community diversity in gestating sows. Comparison of the ARGs and virulence genes carried by them showed that there was a significant positive correlation between the total resistance genome and the total microbiome. However, there were no obvious differences in virulence factors among the different bacterial species.

## Figures and Tables

**Figure 1 vetsci-11-00087-f001:**
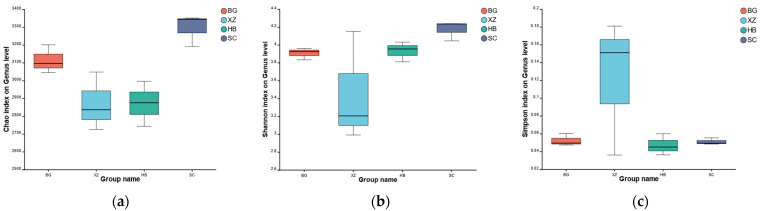
α-Diversity: (**a**) Chao1 index; (**b**) Simpson index; (**c**) Shannon index.

**Figure 2 vetsci-11-00087-f002:**
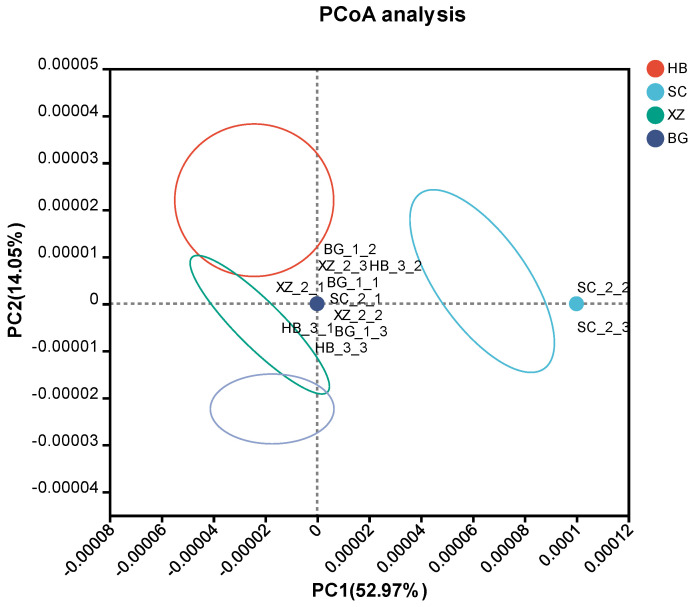
β-diversity. (Figure note: PC1 and PC2 are two principal coordinate components; PC1 represents the principal coordinate component that explains the largest possible change in the data, and PC2 is the principal coordinate component that explains the largest proportion of the remaining degree of change).

**Figure 3 vetsci-11-00087-f003:**
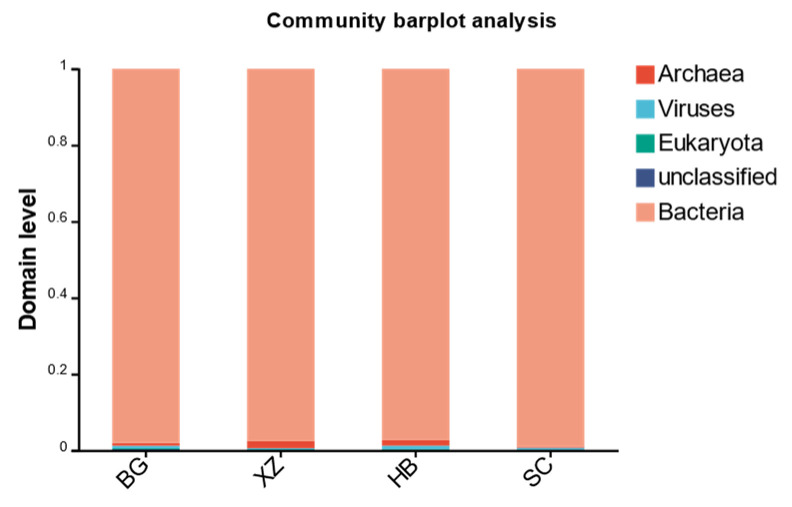
Structural composition of microbial communities.

**Figure 4 vetsci-11-00087-f004:**
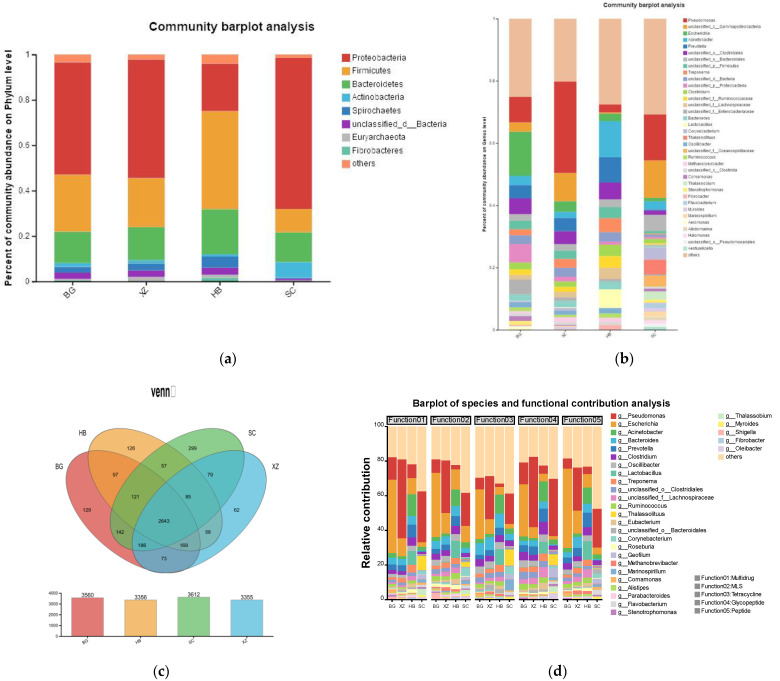
Bacterial community composition. (**a**) Bacterial abundance at phylum level in different samples. (**b**) Genus composition at the genus level. (**c**) Breakdown of the bacterial composition of the four groups of samples. (**d**) Degree of species contribution to ARGs.

**Figure 5 vetsci-11-00087-f005:**
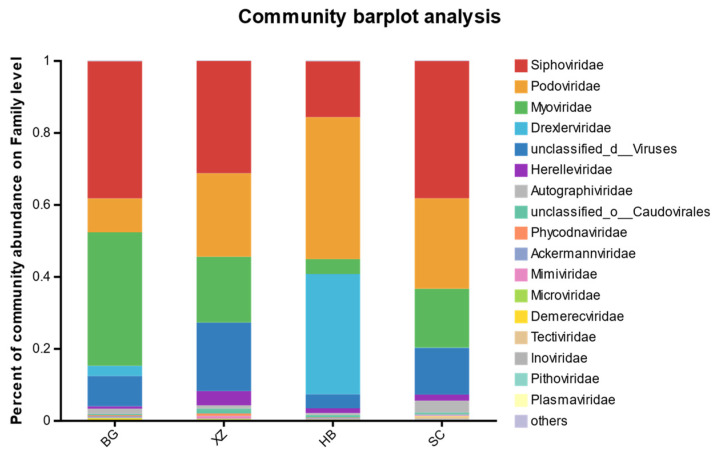
Horizontal structural composition of the virus family.

**Figure 6 vetsci-11-00087-f006:**
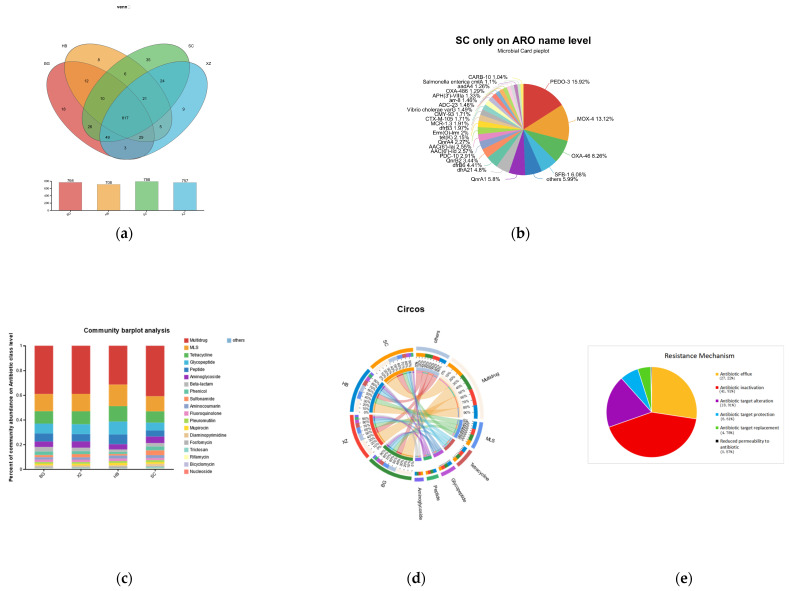
Distribution of antibiotic resistance genes. (**a**). ARG detection in four groups of samples. (**b**). Distribution of resistance genes in SC group. (**c**,**d**). Distribution of different types of antibiotics in each group. (**e**). Main mechanism of action of ARGs in this study sample.

**Figure 7 vetsci-11-00087-f007:**
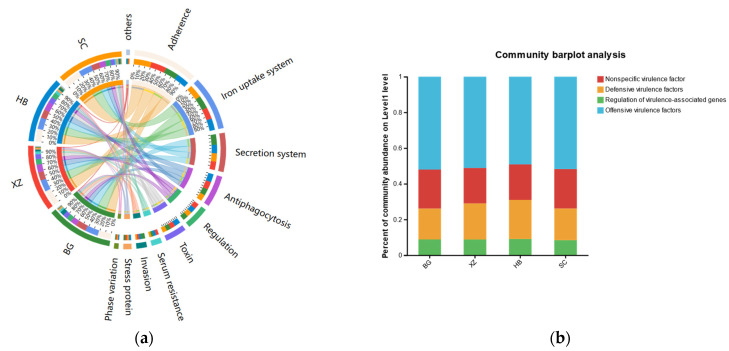
Distribution of virulence factors. (**a**) Distribution of virulence factors by group. (**b**) Percentage of virulence factors by group.

**Table 1 vetsci-11-00087-t001:** Basic information on sequencing.

Sample	Raw Reads	Raw Base (bp)	Contigs	Contigs Bases (bp)	N50 (bp)	N90 (bp)	Max (bp)	Min (bp)	ORFs
BG	44,576,741	6,731,087,941	339,630	236,303,250	737	357	74,979	300	448,655
HB	42,737,859	6,453,416,659	358,414	286,392,037	928	376	177,876	300	478,813
SC	47,829,571	7,222,265,271	336,596	278,158,946	988	374	121,958	300	449,776

Note: Sample: name of the sample, Raw reads: number of sequence entries in raw reads; Raw bases: number of bases in raw reads; Contigs: number of sequence entries in contigs; Contigs bases: total sequence length of contigs; N50 (N90): sort the contigs sequences in order of length, and add up the length values of the scanned sequences one by one from the largest to the smallest, when the cumulative value exceeds 50% (90%) of the total length of all the sequences, then the scanned sequences are summed up. N50 (N90): sort the contigs sequences by their lengths, scan the contigs sequences one by one from the largest to the smallest and add up the lengths of each contigs sequences, when the summed up value exceeds 90% of the total length of all the contigs sequences for the first time, then the scanned sequences will be called N50 (N90); compared with the average length of the contigs sequences, the N50 (N90) is a more accurate representation of the effect of the contigs splicing; Max: the length of the contigs with the longest lengths. Min: length of the shortest contig; ORFs: number of ORFs.

**Table 2 vetsci-11-00087-t002:** Composition of microbial communities.

	Bacteria	Archaea	Viruses	Eukaryota
BG	0.980559	0.007228	0.006983	0.002385
XZ	0.975898	0.018320	0.002967	0.001471
HB	0.973169	0.014486	0.009305	0.002093
SC	0.990551	0.002702	0.004980	0.000697

## Data Availability

All data are contained within the article.

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
