# Peer review of "Metagenome-Based Analysis of the Microbial Community Structure and Drug-Resistance Characteristics of Livestock Feces in Anhui Province, China"

_vetsci, 2024, doi:10.3390/vetsci11020087_

Round 1
Reviewer 1 Report
Comments and Suggestions for Authors
General comment:
The initiate aim of this study is “to obtain information on the distribution of the bacterial flora, virulence genes and ARGs, and their distribution and abundance in the feces of sows during different physiological periods”. However, the achieved results and the conclusion only indicated “there was a significant positive correlation between the total resistance genome and the total microbiome.”, which did not make any comparison and show any differences on the ARGs and virulence genes between groups at four physiological periods.
Since the aim of the study was not supported by the results, and was contradictory with the conclusion, I did not recommended the publication of this article.
The specific problems on each section were list below:
Introduction
1. The length of introduction is not appropriate when compared to other sections. The introduction should focus on the significance of the study on ARGs and virulence genes, with concise descriptions;
2. The advantages and the reason for using metagenomic sequencing should be thoroughly stated, such as higher level of phylotype resolution when compared to 16S rRNA gene sequencing;
Reference on regional investigation such as https://doi.org/10.1016/j.jhazmat.2022.129719. may help the author better understanding the description.
Materials and Methods
Line 127: “No fewer than three samples (each no less than 10 g) were collected for each period.” . The number of pig samples was too small and resulted in big differences between individuals within groups, especially Group XZ, which showed by Contigs and ORFs in Table 1, and by extremely high bars in Figure 1 in section 3.1. Accordingly, the result from Group XZ was not convincing.
Results
1. The data in Table 1 should be rearranged, values in the same group can be assembled;
2. The meaning of CHAO1, Simpson, and Shannon should be changed to section “Materials and Methods”;
3. Figure 1 and 2 are not cited in the main manuscript;
4. The data point of each individual sample was not exhibited in Figure 2 but only circles?
5. I recommend using table to replace Figure 3 due to invisibility of fungi.
6. The ratio and resolution of Figure 4 should be readjusted, species data should be provided in Figure 4, and the legend of Figure 4d is incomplete and incorrect.
Discussion
1. Paragraph 1: As shown, “The main objective of this study was to characterize the functional content of the porcine fecal microbiome.” However, this paragraph only discussed the bacterial diversity differences between groups and potential reasons.
2. Paragraph 2: “The antibiotic inactivation mechanism was the primary mechanism of action for the development of resistance in the samples of this study, followed by the antibiotic efflux system.” This conclusion was not supported by the achieved results in current work. As shown in Figure 4, the data only showed the most common mechanism in all groups but could not demonstrate the development of resistance in the sows during different physiological periods.
3. Paragraph 3: Line 379, “Notably, the authors of a previous study found a significant positive correlation between the total resistance genome and the total microbiome, [30].” Accordingly, this statement is suitable to be used as the major conclusion at line 397 and 398 for this study.
Comments on the Quality of English LanguageMinor editing of English language required.
Author Response
Introduction
- The length of introduction is not appropriate when compared to other sections. The introduction should focus on the significance of the study on ARGs and virulence genes, with concise descriptions;
Reasonable deletion of the introduction and highlighting the significance of the study of drug resistance genes and virulence genes.
- The advantages and the reason for using metagenomic sequencing should be thoroughly stated, such as higher level of phylotype resolution when compared to 16S rRNA gene sequencing;
Added the advantages of macro-genome sequencing, and made a comparison with 16S rRNA sequencing technology, highlighting the specific advantages of macro-genome sequencing.
Materials and Methods
Line 127: “No fewer than three samples (each no less than 10 g) were collected for each period.” . The number of pig samples was too small and resulted in big differences between individuals within groups, especially Group XZ, which showed by Contigs and ORFs in Table 1, and by extremely high bars in Figure 1 in section 3.1. Accordingly, the result from Group XZ was not convincing.
Thank you very much for being able to make suggestions, but we were unable to revise it in time because the article had already been molded and it was not possible to reintegrate the data. Next time we will definitely consider the relevant factors in advance to avoid the same problem.
Results
- The data in Table 1 should be rearranged, values in the same group can be assembled;
Changes have been made to article-related data.
- The meaning of CHAO1, Simpson, and Shannon should be changed to section “Materials and Methods”;
I apologize because the focus of the "Materials and Methods" section is on the experimental process, and putting the meaning of CHAO1, Simpson, and Shannon here is slightly stiff, so I have not made any changes. - Figure 1 and 2 are not cited in the main manuscript;
Figures 1 and 2 have been added as references in the main manuscript - The data point of each individual sample was not exhibited in Figure 2 but only circles?
In Figure 2, since the sample points are too few, we do not show the sample points but choose to use confidence circles instead. - I recommend using table to replace Figure 3 due to invisibility of fungi.
Thank you for your advice.Table 2 has been added.
- The ratio and resolution of Figure 4 should be readjusted, species data should be provided in Figure 4, and the legend of Figure 4d is incomplete and incorrect.
Images in the text have been provided as vectors for viewing .
Discussion
- Paragraph 1: As shown, “The main objective of this study was to characterize the functional content of the porcine fecal microbiome.” However, this paragraph only discussed the bacterial diversity differences between groups and potential reasons.
Because this study focuses on the analysis and discussion of the functional content of the porcine fecal microbiome from the point of view of fecal microbial diversity, as the first paragraph of the discussion we analyzed the samples in terms of the differences in bacterial diversity between the groups.
- Paragraph 2: “The antibiotic inactivation mechanism was the primary mechanism of action for the development of resistance in the samples of this study, followed by the antibiotic efflux system.” This conclusion was not supported by the achieved results in current work. As shown in Figure 4, the data only showed the most common mechanism in all groups but could not demonstrate the development of resistance in the sows during different physiological periods.
Figure 6e supports the conclusion that "antibiotic inactivation was the main mechanism of action for the development of resistance in the samples of this study, followed by the antibiotic efflux system". The intergroup variability was discussed in terms of microbial composition, and according to the data analyzed in this study, there was a direct correlation between drug resistance and bacterial species, which in turn was related to the type of resistance.
- 第 3 段:第 379 行,“值得注意的是,先前研究的作者发现总抗性基因组和总微生物组之间存在显着的正相关关系,[30]。因此,该陈述适合用作本研究第 397 行和第 398 行的主要结论。
感谢您的建议,已根据您的要求将句子移回。

Reviewer 2 Report
Comments and Suggestions for Authors
The overall design of the study is complete and well written. The results of the study are of some practical value as they shed light on and promote practical farming. However, the manuscript needs to be slightly improved, and there are some issues that need to be carefully considered:
1. The presentation of the introduction part of the article is cumbersome, and it is recommended to streamline the relevant content to make the presentation more concise. There are grammatical problems in the introduction, which need to be revised to improve the accuracy of the presentation.
2. The authors did not clearly explain the source of the samples in the article, and it is hoped that relevant details can be added to enhance the transparency of the experimental design.
3. In the Materials and Methods section of the article, the software involved should be uniformly given a version number to ensure the consistency of the format. The authors are requested to check whether there is an error in the grouping expression in line 211. In addition, the HS group is not mentioned in the article, and it is suggested to make additions.
4. For the use of acronyms, the full name should be given at the first occurrence. Please check the full text. In addition, the initial letters of polymerase chain reaction (PCR) in line 56 should be capitalized.
5. In line 40, “drug resistance genes (ARGs)”, it should be “antibiotic-resistance genes (ARGs)”. Please confirm this.
Author Response
- The presentation of the introduction part of the article is cumbersome, and it is recommended to streamline the relevant content to make the presentation more concise. There are grammatical problems in the introduction, which need to be revised to improve the accuracy of the presentation.
The introductory section of the article has been revised based on expert comments and is available for review.
- The authors did not clearly explain the source of the samples in the article, and it is hoped that relevant details can be added to enhance the transparency of the experimental design.
At the request of the farm, because of the need to protect its relevant information, so the text to a large-scale farm on behalf of the name, I hope you can understand.
- In the Materials and Methods section of the article, the software involved should be uniformly given a version number to ensure the consistency of the format. The authors are requested to check whether there is an error in the grouping expression in line 211. In addition, the HS group is not mentioned in the article, and it is suggested to make additions.
I apologize for my mistake, the article wanted to express as HB group. Also have added the appropriate version number.
- For the use of acronyms, the full name should be given at the first occurrence. Please check the full text. In addition, the initial letters of polymerase chain reaction (PCR) in line 56 should be capitalized.
感谢您的提醒,我们已经检查了全文。
- 在第 40 行“耐药基因 (ARG)”中,应为“抗生素耐药基因 (ARGs)”。请确认这一点。
非常感谢您的建议,我会修改它。

Reviewer 3 Report
Comments and Suggestions for Authors
Shao et al here reports that microbial community structure and drug resistance characteristics of livestock and poultry feces in Anhui province, China by using metagenome-based analysis. They found the distribution and abundance of the bacterial flora, virulence genes and antibiotic-resistance genes in the feces of sows during different physiological periods. The study does not lack novelty and the conclusion is convinced.
Author Response
Thank you very much for your affirmation of the content of the article.
Reviewer 4 Report
Comments and Suggestions for Authors
The authors have studied porcine gut microbiome from different stages of sows at a pig farm. Study is limited by number of samples and hence, should be considered a pilot.
Title: Why are poultry mentioned here? It should be strictly swine or porcine.
Intro: line 43-44-"Pathogenic ARGs". Genes are not pathogenic on their own. Perhaps you are referring to bacterial pathogens. Consider correcting or removing.
Line 67: "tuberculosis bacilli" are you referring to Mycobacterium?
Methods: The sampling strategy needs to be defined better. What are these periods? Can you define them? Are the samples taken from individual pigs or pooled ? Here, it is mentioned that one farm was studied, whereas in abstract multiple farms are mentioned. What was the management on the farm? Were antibiotics used?
Section 2.5: Explain what CARD and VFDB are for the readers. What settings were used to run blast these databases? Please expand the bioinformatic analysis elsewhere too.
Figures in general are extremely hard to read, and must be magnified.
Discussion: Please comment on how these results match with those presented in other studies globally.
Author Response
Title: Why are poultry mentioned here? It should be strictly swine or porcine.
Thanks for the heads up, the title has been redacted.
Intro: line 43-44-"Pathogenic ARGs". Genes are not pathogenic on their own. Perhaps you are referring to bacterial pathogens. Consider correcting or removing.
The relevant statement has been modified.
Line 67: "tuberculosis bacilli" are you referring to Mycobacterium?
Yes, when we refer to the tuberculosis bacillus, we are referring to Mycobacterium tuberculosis. This bacterium is the causative agent of tuberculosis, a severe respiratory disease.
Methods: The sampling strategy needs to be defined better. What are these periods? Can you define them? Are the samples taken from individual pigs or pooled ? Here, it is mentioned that one farm was studied, whereas in abstract multiple farms are mentioned. What was the management on the farm? Were antibiotics used?
Our samples were collected from a large-scale farm, where the agricultural products were managed according to their internal regulations and the animals were not treated with antibiotics when they were healthy.
Samples were collected from a mix of pigs in the same pen and then subjected to high-throughput sequencing, in different physiological periods.
- Reserve period: also known as the recovery period, this is the phase of recovery after the animal has given birth, during which the female needs to replenish the nutrients lost during gestation and lactation.
- Vacancy: This is the stage when animals are sexually dormant and females are not sexually active between breeding seasons.
- Gestation: Gestation is the period from conception to delivery, which is the period during which an animal's fetus develops in the mother's body.
- Mating period: This is the stage when the female animal receives the male animal for mating, which is often indicated by certain physiological and behavioral changes, such as estrus in many mammals.
Section 2.5: Explain what CARD and VFDB are for the readers. What settings were used to run blast these databases? Please expand the bioinformatic analysis elsewhere too.
Figures in general are extremely hard to read, and must be magnified.
In this study our thresholds were all set to 80%.
Discussion: Please comment on how these results match with those presented in other studies globally.
Relevant studies have been cited in the text; thanks for reviewing.

Round 2
Reviewer 4 Report
Comments and Suggestions for Authors
Thank you, but not all of my concerns have been satisfied.
1. Add the information on farm management in the article, not just in response to reviewer
2. Spell CARD and VFDB
3. Figures are still not clear
Author Response
1. Add the information on farm management in the article, not just in response to reviewer
Relevant information has been added to the text.
This is a group of large-scale pig farms, with our group is a long-term cooperation, due to the requirements of the internal management of the group company, breeding information (immunization program, feed, morbidity and medication, etc.) involves the privacy of the farm, the farm can not disclose the above information.
If we need the farms to provide the proof of not being able to disclose the management mode of the farms, we can provide the scanned copy of the proof with the seal in the future.
2. Spell CARD and VFDB
Relevant spellings have been added to the text.
3. Figures are still not clear
The original drawings have been submitted with attachments for your review.
